# Solving Coastal Dynamics: Introduction to High Resolution Ocean Forecasting Services

Joanna Staneva[1], Angelique Melet[2], Jennifer Veitch[3], Pascal Matte[4]

[1]Institute of Coastal Systems - Analysis and Modelling, Helmholtz-Zentrum Hereon, Geesthacht, Germany
[2]Mercator Ocean International, Toulouse, France
[3]Egagasini Node, South African Environmental Observation Network (SAEON), Cape Town, South Africa
[4]Meteorological Research Division, Environment and Climate Change Canada, Québec, QC, Canada

*Correspondence to*: Joanna Staneva (joanna.staneva@hereon.de)

**Abstract.** Coastal services are fundamental for society, with approximately 60% of the world's population living within 60 km of the coast. Thus, predicting ocean variables with high accuracy is a challenge that requires numerical models able to simulate from mesoscale to submesoscale processes, to capture shallow water dynamics influenced by wetting-drying and resolve the ocean variables in very high-resolution spatial domains. This paper introduces key aspects of coastal modelling, such as vertical structure of the mixed layer depth, parameterization of bottom roughness and the dissipation of kinetic energy in coastal areas. It stresses the need for models to account for the nonlinear interactions between tidal currents, wind waves, and small-scale weather patterns, emphasizing their significance in refining coastal predictions. In addition, observational advancements, such as high-frequency (HF) radar and satellite missions like SWOT, provide unique opportunities to observe coastal dynamics. This integration enhances our ability to model physical and dynamical peculiarities in coastal waters, estuaries, and ports. Coastal models not only benefit from such high-resolution observations but also contribute to evolving observational systems, creating feedback loops that refine monitoring and prediction capabilities. Modelling strategies are also examined, including downscaling and upscaling approaches, and numerical challenges like implementing robust data assimilation schemes to refine estimations of coastal ocean states are addressed. Emerging techniques, such as advanced turbulence closure models and dynamic vegetation drag parameterization, are highlighted for their role in enhancing the realism of modeled coastal processes. Furthermore, the integration of atmospheric forcing, tidal asymmetries, and estuarine dynamics underlines the necessity for models that span the complexities of the coastal continuum. It also demonstrates the critical importance of accurately modelling coastal and estuarine systems to capture interactions between mesoscale and submesoscale processes, their connections to broader oceanic systems, and their implications for sustainable coastal management and climate resilience. This work underscores the potential of advancing coastal forecasting systems through interdisciplinary innovation, paving the way for enhanced scientific understanding and practical applications.

## 1 Introduction

High resolution observation and modelling are needed so that marine services can be compliant with small-scale processes in the ocean, particularly in coastal areas where these processes have a significant impact on dynamics and biogeochemistry (Figure 1). The importance of high resolution in coastal services is underscored by the coastal ocean's significance to humanity, not least because about 60% of the world's population lives within 60 km of the coast (Rao et al., 2008). These areas are highly dynamic, subject to both direct and indirect anthropogenic impacts, respectively, such as eutrophication, overfishing, offshore wind farm development, dredging, and pollution, global warming, sea level rise and changes in meteorological and hydrological conditions. These combined influences frequently trigger regime shifts, coastal erosion, flooding, and the introduction of invasive species, underscoring the vulnerability and complexity of these systems.

Accurately predicting ocean variables in coastal environments is challenging due to the need to resolve mesoscale to submesoscale dynamics and their interactions with atmospheric and hydrological processes. The inherent variability of these systems requires models that can account for a wide range of phenomena, including tidal asymmetries, wetting-drying cycles, nonstationary river and atmospheric forcing, and nonlinear feedback mechanisms between tidal currents and wind waves (Staneva et al., 2017). These processes influence mixing, ocean circulation, and the accuracy of sea surface temperature predictions. Thus, high-resolution models are indispensable for capturing the fine-scale interactions that drive coastal dynamics and shape biogeochemical responses.

Observational data play a pivotal role in advancing coastal modelling. High-frequency (HF) radar and novel high resolution satellite missions offer unprecedented opportunities to observe and understand coastal processes with fine spatial and temporal resolution (De Mey-Frémaux et al., 2019). These data sources are integral to improving the representation of physical and biogeochemical variability in the models, bridging the gap between observations and predictive frameworks. By integrating data from remote sensing and in situ platforms, coupled with advanced data assimilation techniques, models can better capture the complexity of estuarine and nearshore processes.

Changes occurring in the coastal ocean are attributed to both direct human impacts and climate change. Anthropogenic impacts encompass factors such as eutrophication, overfishing, offshore wind farm construction, dredging, and pollution. Natural changes in the coastal ocean result from sea-level rise, global warming, and alterations in meteorological and hydrological conditions such as precipitation, evaporation, wind patterns, and river run-off. These natural and human-induced changes can lead to significant regime shifts, including alterations in biogeochemistry, increased coastal erosion, heightened flooding risks, and the proliferation of invasive species, among other impacts.

Science-based services in the coastal ocean are essential for ensuring efficient management, sustainable use of coastal systems, and the development of strategies that are adaptable to the changing climate, including sea-level rise. These efforts, for example, align with the marine strategy framework directive in the European context (Hyder et al., 2015).

The aim of this paper is to introduce high-resolution ocean forecasting services that address the challenges of coastal dynamics by improving predictions of physical and biogeochemical processes. It focuses on the integration of advanced modelling

techniques and modern observational tools to enhance understanding of small-scale dynamics and their connections to larger ocean systems. The paper first describes the spatial scales and processes that high-resolution models address, focusing on local, regional, and transitional zones. It then explores advanced observational tools, such as satellite missions and HF radars, and their role in improving coastal forecasts. Following this, the discussion highlights numerical modelling techniques, including turbulence modelling and bottom drag parameterization, which are essential for capturing small-scale coastal dynamics. It also examines the role of data assimilation techniques and observing system experiments in improving prediction accuracy and guiding the design of observation networks. Finally, the paper concludes with a summary of findings, identifies current challenges, and outlines future directions for advancing coastal forecasting systems. By addressing these topics, the paper aims to support the development of more robust and adaptable tools for coastal forecasting, which are critical for sustainable management and improving resilience to environmental changes.

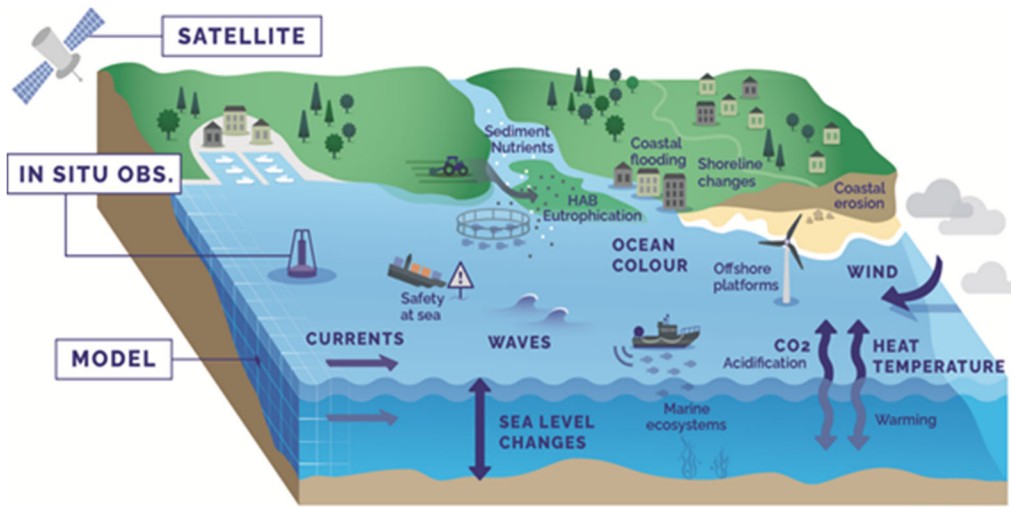

**Figure 1: Schematic representation of the coastal zone, hazards (e.g. HAB (harmful algea bloom), metocean and biogeochemical variables, as well as observations and applications (adapted from Melet et al., 2020).**

## 2 Typical spatial scales and processes solved by high-resolution services

High-resolution services in the coastal ocean operate at various spatial scales depending on the specific applications and objectives. These scales can range from local to regional levels, aiming to capture fine-scale processes and variations. Here are some typical spatial scales for high-resolution services:

*1. Local Scale:* At the local scale, high-resolution services focus on small coastal areas, such as individual bays, estuaries, or nearshore zones. These services aim to provide detailed information and predictions for specific locations of interest. Spatial

resolutions in this range can be on the order of meters to a few kilometers, allowing for precise observations and modelling of
localized processes.
*2. Coastal Scale:* High-resolution services at the coastal scale cover larger coastal regions, spanning multiple bays, estuaries,
and coastal zones. These services provide a broader view of the coastal environment and its dynamics. Spatial resolutions in
this range typically range from meters to a kilometer, enabling the capture of coastal- to regional-scale variations and
interactions.
*3. Transition Zones:* Transition zones refer to areas where coastal and open ocean processes interact. These zones often exhibit
complex dynamics and are of particular interest for high-resolution services. Spatial resolutions in transition zones can vary
depending on the specific characteristics and objectives, but they generally aim to capture the intricate interactions between
coastal and open ocean processes.
A collection of 11 recent studies on operational coastal services utilizing high-resolution models offers significant insights into
the relevant spatial scales, objectives, and applications, thereby strengthening the analysis in this context (Sotillo, 2022). Eddies
or isolated vortices, meandering currents or fronts and filaments are characteristic features of oceanic mesoscale processes.
These processes typically exhibit spatial scales ranging from 10 to 500 kilometers, depending on geographic latitude and
stratification, and time scales ranging from several days to approximately 100 days. Submesoscale processes in the ocean, on
the other hand, are characterized by smaller scales, typically ranging from 1 to 10 kilometers (McWilliams, 2016). These scales
are smaller than the Rossby radius of deformation. Submesoscale processes also have shorter temporal scales, usually lasting
only a few hours, and their relative vorticity is greater than the Coriolis parameter f. In contrast, for mesoscale motion, the
relative vorticity is comparable to f. Overall, studying and observing submesoscale processes require advanced techniques and
methods to overcome their small scale and rapid variability, but their understanding is crucial for comprehending the intricate
dynamics of the ocean.
The surface and bottom mixed layers in the open ocean occupy just a tiny part of the ocean volume because these layers are
much thinner than the almost viscousless ocean interior. However, in the coastal zone, drag parameterizations become
increasingly important in shallow water, and even more so where the impact of vegetation is significant. Furthermore, a large
part of kinetic energy in the ocean is dissipated in the coastal zone, which necessitates an adequate modelling of this important
small-scale process, vital for the global energy balance (Munk and Wunsch, 1998). To accurately represent the coastal
dynamics and the fine structure of these layers, models need to resolve the vertical structure of the mixed layers. This
requirement necessitates the use of turbulence closure models, which account for the effects of turbulence and mixing in these
regions. Additionally, models for coastal processes need to consider the impact of bottom drag. The parameterization of bottom
roughness, often based on the grain size distribution, allows for the inclusion of bottom drag effects. In cases where vegetation
is present, drag parameterizations become even more important. A significant portion of the kinetic energy in the ocean is
dissipated in the coastal zone. Therefore, it is crucial to adequately model these small-scale processes in order to maintain a
balanced representation of the global energy dynamics. Understanding and accurately simulating the dissipation of kinetic
energy in coastal areas contribute to a comprehensive understanding of the ocean's energy budget.

In shallow water, the variability of surface elevation caused by tides and storms becomes comparable to the water depth itself. In some coastal areas, shallow-water tides play a significant role in the overall tidal dynamics. To improve the accuracy of tidal predictions in shelf regions, it is necessary to consider higher harmonics and assess the ability of ocean models to fully resolve the tidal spectrum.

Some important processes, such as the nonlinear feedback between strong tidal currents and wind waves, cannot be ignored in the coastal zone (Staneva et al., 2016a, 2016b, 2017). Wave-current coupling tends to decrease strong winds through wave-dependent surface roughness (Wahle et al., 2017), affects mixing and ocean circulation, and improves predictions for sea surface temperature. Further examples of the value of the incorporation of coupling in the numerical models in the coastal ocean are given by De Mey-Frémaux et al. (2019). These scientific developments of operational oceanography are in pace with the trend in the Earth System modelling to seamlessly couple different environmental prediction components of atmosphere, waves, hydrology, and ice.

The small spatial scales characteristic of coastal and estuarine systems requires coastal models to consider ageostrophic (deviating from the Earth's rotation) and three-dimensional dynamics, primarily driven by boundary-layer processes (Fringer et al., 2019). Understanding these small-scale processes is crucial, particularly the interactions between mesoscale and sub-mesoscale dynamics and their connection to larger-scale processes. It is essential to improve the representation of exchanges between the coastal and open ocean, as well as their coupling with estuaries and catchment areas, in order to capture the complexity of coastal systems. Accounting for high-resolution atmospheric forcing in the coastal models is essential for accurately capturing local meteorological dynamics, including wind patterns, temperature gradients, and precipitation rates. Such detailed atmospheric data drive fundamental processes like heat and momentum fluxes, profoundly influencing coastal hydrodynamics, sediment transport, and ecosystem responses. The implementation of a novel high-resolution atmospheric forcing, combined with the refinement of bulk formulae for surface flux computations, significantly enhances the performance of various high-resolution modelling systems for port environments (García-León et al.). Coastal models need to accurately account for frictional balances, taking into consideration the effects of friction on the movement of water. They must also address wetting and drying processes, as well as hydrological forcing, to capture the transitions between shallow environments and larger regional scales. By incorporating these factors, models can provide a more realistic representation of coastal dynamics. In addition, the grid characteristics used in coastal models should be carefully selected to accurately represent the dominant spatial scales present in the coastal environment. Choosing grid resolutions that capture the essential features of the coastal system is crucial for obtaining reliable and meaningful results.

In the coastal ocean, characteristic time scales are significantly shorter compared to the global ocean. These time scales, typically around 1 day, are determined by various processes, including tides, inertial motion, diurnal cycles, and synoptic weather patterns. The fast-paced dynamics of the coastal ocean require models to accurately capture these shorter time scales. In estuaries, the periodicity becomes more complex due to strong tidal asymmetries and the presence of secondary circulation patterns. The interactions between tidal forcing, river flow and estuarine geometry result in intricate and variable periodic patterns. (as shown in Campuzano et al. 2022 for the Western Iberian Buoyant Plume, Sotillo et al. 2021 for the whole European

Atlantic façade, Pein et al. 2021 for the Elbe Estuary). The periodicity observed in coastal seas is mainly influenced by external forcing signals, such as atmospheric conditions or remote ocean signals. These external signals propagate in the coastal models through the specification of lateral boundary conditions, which is a crucial aspect of modelling in coastal areas. Unlike global models that can operate without open boundaries, coastal models require careful consideration of these boundary conditions to accurately represent the interactions between the coastal and open ocean.

The predictability limit of models depends on the geophysical processes. For synoptic processes in the open ocean, this limit is on the order of weeks to months. For the coastal ocean, it is on the order of hours to days. The loss of predictability, associated with nonlinear processes, is exemplified by the growth of errors in predictive models. Assimilation of data containing spatial and temporal scales below the predictability limit is needed to address this issue. Simulations at grid resolutions that would sufficiently resolve the coastal submesoscale would require horizontal grid resolutions of approximately 1-10 meters in estuaries and 0.1-1 kilometer in coastal shelf domains. However, achieving such high resolutions poses significant computational challenges and resource demands.

By employing high-resolution services with appropriate spatial scales, scientists and stakeholders can gain a more detailed and accurate understanding of coastal processes, improve forecasting capabilities, and support effective coastal management and decision-making.

## 3 State-of-the-art data and tools for coastal forecasting

### 3.1 Required observations

Observing systems are spatiotemporally sparse in coastal regions compared to the small scales of ecosystem variability found there. A crucial challenge in observations is addressing the variety of important spatial and temporal scales within the coastal continuum, which encompasses the seamless transition from the deep ocean to estuaries through the shelf. In order to achieve this, observations should sample the multiscale, two-way interactions of estuarine, nearshore, and shelf processes with open ocean processes. Additionally, they need to account for the different pace of circulation drivers, such as fast atmospheric and tidal processes, as well as the slower general ocean circulation and climate forcing. It is also important to accurately sample the gradients of biological production, ranging from mesotrophic estuaries to oligotrophic oceans. Given the current situation, observational practices and strategies need to be strongly coupled with numerical modelling to effectively extract the information contained in the data and advance the quality of coastal services.

Most global and regional prediction products use a combination of satellite observations and in situ observations. Traditionally, in situ observations constituted the major data source for coastal ocean monitoring. During the end of the past century, satellite observations contributed significantly to the understanding of spatial variabilities. Novel instruments, such as the acoustic Doppler current profiler (ADCP), which measures current profiles throughout the water column, enhanced our understanding of current shear and bottom stress. Nowadays, high-resolution numerical simulations in the coastal ocean are keeping pace with high-resolution observations. A similar trend is observed in coastal waters, estuaries, and ports, which are rich in different

activities and interests: fishing, recreational activities, search and rescue, protection of habitats, storm forecasts, maritime industries, as well as routine maintenance operations (De Mey-Frémaux et al., 2019).

The coastal ocean observations only are not sufficient to fully support the present-day need for high-quality ocean forecasting and monitoring because measurements may represent very localized and short scale dynamics, and it is not straightforward to know how fully they describe the complex coastal system. Therefore, recent practices employ the synergy between observations and numerical modelling, which ensures valuable research advancements and practical implementations (Kourafalou et al., 2015a, 2015b). The core components of operational oceanographic systems consist of a multi-platform observation network, a data management system, a data assimilative prediction system, and a dissemination/accessibility system (Kourafalou et al., 2015a; De Mey-Frémaux et al., 2019; Davidson et al., 2019). By combining observations and models through data assimilation methods, ranging from coastal to global and from in situ to satellite-based, we can assess ocean conditions and create reliable forecasts. This integration adds value to coastal observations and enables a wide range of applications (De Mey-Frémaux et al., 2019; Ponte et al., 2019), as well as providing decision-making support. For a comprehensive review of ocean monitoring and forecasting activities in both the open and coastal oceans, please refer to Siddorn et al. (2016).

High-frequency radars (HFR) offer unique spatial resolution by providing reliable directional wave information and gridded data of surface currents in almost real time. The use of HFR networks has become an essential element of coastal ocean observing systems, contributing to high-level coastal services (Stanev et al., 2016; Rubio et al., 2017; Reyes et al., 2022). The outputs from prediction systems extend the utility of HFR observations beyond the immediate observation area (Stanev et al., 2015), enabling adequate estimates even where no direct observations have been made. This demonstrates how models connect observations, synthesize them, and assist in the design of observational networks. In turn, observations can guide the development of coastal models (De Mey-Frémaux et al., 2019).

Alongside ADCP data, HFR data are used for skill assessment of operational wave and circulation models (Lorente et al., 2016). Another valuable source of fine-resolution data in the coastal region is provided by color data from satellites. In terms of sea level observations, some challenges associated with the use of altimeter data in the coastal zone are expected to be overcome through the use of wide-swath Surface Water and Ocean Topography (SWOT) technology. SWOT is a landmark satellite mission that delivers two-dimensional sea surface height observations at high resolution across a 120 km swath. It represents a major step forward in resolving mesoscale and submesoscale features critical to coastal dynamics. Recent Observing System Simulation Experiments (OSSEs) have demonstrated that wide-swath altimetry substantially enhances ocean forecasting capabilities. For instance, a constellation of two SWOT-like wide-swath altimeters provides a ~14% reduction in sea surface height forecast error compared to a 12-nadir altimeter constellation and also improves estimates of surface currents and Lagrangian trajectories (Benkiran et al., 2024). These results highlight the importance of SWOT-type observations for resolving small-scale coastal variability and improving model-data integration.

Further advances in coastal observations are enabled by autonomous platforms such as Slocum gliders. These gliders can carry a wide array of physical and biogeochemical sensors and perform repeated transects, thus providing high-resolution

observations of dynamic features such as eddies, frontal systems, and upwelling events. Their operational flexibility and ability to collect subsurface data make them valuable for both sustained monitoring and adaptive sampling strategies (Rudnick, 2016; Testor et al., 2019). In parallel, satellite technologies continue to evolve. Moreover, the Japanese geostationary meteorological satellite Himawari-8 provides high-frequency (every 10 minutes) and high-resolution (up to 500 m) visible and infrared imagery. These capabilities allow for near-real-time monitoring of sea surface temperature (SST), making it possible to track rapidly evolving coastal phenomena such as diurnal warming, river plumes, and thermal fronts (Kurihara et al., 2016). These complementary in situ and remote sensing platforms represent essential components of integrated coastal observing systems, supporting the growing demand for accurate forecasts, early warnings, and data-driven decision-making tools.

## 3.2 Numerical models

Addressing specific processes in the coastal ocean and accurately modelling the transition between regional and coastal scales cannot be achieved solely by adjusting the model resolution. Certain processes, such as shallow-water tides, which are often overlooked in global and regional forecasting, play a dominant role in coastal ocean dynamics. The previous sections have highlighted the importance of a tailored approach in observational practices and numerical models for the coastal ocean. For further information on other popular coastal models, refer to the comprehensive discussion by Fringer et al. (2019).

**Table 1: Circulation models in alphabetical order, which can be used for coastal and regional studies and/or provision of services.**

| Model | Citationan | C: Coastal, R: Regional, G: Global | Finte-volume (FV) or Finite-element (FE) |
|---|---|---|---|
| ADCIRC | Luettich et al. (1992) Westerink et al. (1994) | C | FE |
| COAWST | Warner et al. (2008, 2010) | C/R | FV |
| COMPAS | Herzfeld et al. (2020) | C/R | FV |
| CROCO | Marchesiello et al. (2021) | C/R | FV |
| Delft3D | Delft3D-Flow User Manual (2024) | C | FV |
| FVCOM | Chen et al. (2003) | C/R/G | FV |
| GETM | Burchard and Bolding (2002) | C | FV |
| MITgcm | Marshall et al. (1997) | C/R/G | FV |
| MPAS | Ringler et al. (2013) | R/G | FV |

| | | | |
|---|---|---|---|
| NEMO | Madec et al., (2016) | C/R/G | FV |
| POMS | Blumberg and Mellor (1987), Mellor (2004) | C/R | FV |
| ROMS | Shchepetkin and McWilliams (2005) | R | FV |
| SCHISM | Zhang et al. (2016) | C/R/G | FV/FE |
| SELFE | Zhang and Baptista, 2008 | C | FV/FE |
| SHYFEM | Umgiesser et al. (2004) | C | FE |
| SUNTANS | Fringer et al. (2006) | C | FV |
| TRIM/UnTRIM | Casulli (1999), Casulli and Zanolli (2002, 2005) | C | FV |

**3.3 Fine resolution nested models, downscaling and upscaling**

High-resolution coastal services must properly resolve interactions between various coastal processes, including nearshore, estuarine, shelf, drying, and flooding dynamics. Achieving this requires a resolution of approximately 10-100 meters. Simultaneously, it is essential to capture open ocean processes at a resolution of around 1 kilometer or coarser. Common approaches employed in addressing this challenge include downscaling and multi-nesting techniques (e.g., Debreu et al., 2012; Kourafalou et al., 2015b; Trotta et al., 2017) as well as the use of unstructured-grid models (e.g., Zhang et al., 2016a, 2016b; Federico et al., 2017; Stanev et al., 2017; Ferrarin et al., 2018; Maicu et al., 2018). Another important aspect to consider is upscaling (Schulz-Stellenfleth and Stanev, 2016), which becomes relevant when addressing the two-way interaction between coastal and open-ocean systems.

Most coastal models are one-way nested, relying heavily on forcing data from larger-scale models as the coastal system is primarily influenced by the atmosphere, the hydrology and the open ocean. Enhancing the horizontal resolution of the North Sea operational model from 7 to 1.5 kilometers (Tonani et al., 2019) has shown improvements in off-shelf regions, but biases persist over the shelf area, indicating the need for further enhancements in surface forcing, vertical mixing, and light attenuation.

An important consideration in downscaling and coastal modelling is the treatment of open boundary conditions (OBCs), which play a critical role in determining model fidelity near the boundaries. OBCs are typically derived from larger-scale models but often require case-specific tuning to ensure dynamic consistency and minimize reflection or spurious signals. The choice and configuration of OBCs—such as Flather-type, radiation conditions, or relaxation zones—can significantly affect the transport and energy balance within the coastal model domain. Given the diversity of physical processes and geometries encountered in

coastal environments (Marchesiello et al., 2001). Models equipped with a wide suite of configurable boundary condition types offer a practical advantage, particularly in multi-scale coupled frameworks. Ensuring consistency across nested domains while preserving physical realism remains an ongoing challenge, motivating continued development and intercomparison of OBC strategies in operational and research settings.

While the downscaling of information from coarser global or regional models to high-resolution coastal models is well-established, the reverse process of upscaling is more challenging and continues to be a subject of research. Two-way nested models allow assimilated information from coastal observations, typically not assimilated by larger-scale forecasting systems, to propagate beyond the coastal region while maintaining dynamic consistency. This upscaling capability has the potential to benefit regional models. Coastal observations have demonstrated their potential to improve boundary forcing or surface wind forcing in regional models.

The coupling of a coarse-resolution regional model with a fine-resolution coastal model using a two-way nesting approach has been studied in the context of the straits connecting the North and Baltic Seas. The intricate topography and narrow cross-sections of the straits result in the dominance of small-scale motions, which play a vital role in the exchange between the two seas and significantly influence Baltic Sea stratification. The two-way nesting method, design to exchange information between the child model in the straits and the parent model in the seas, incorporates elements of data assimilation and allows for different vertical discretizations in each model. The Adaptive Grid Refinement in FORTRAN (AGRIF), originally developed by Debreu et al. (2008; 2012), has found wide application as a library for seamless spatial and temporal refinement over rectangular regions in the NEMO modelling framework (https://forge.ipsl.jussieu.fr/nemo/wiki/WorkingGroups/AGRIF). Recent advancements in two-way nesting frameworks have demonstrated their effectiveness in improving multi-scale model accuracy. The implementation of a general two-way nesting framework has enhanced the exchange of physical properties between nested grids while preserving numerical stability and computational efficiency. Additionally, the integration of two-way nesting in a global ocean model has significantly improved surface tidal accuracy, refining regional tidal dynamics without compromising large-scale coherence (Herzfeld & Rizwi, 2019; Jeon et al., 2019). Further applications of AGRIF have demonstrated improvements in hydrodynamic simulations and the estimation of environmental indicators in coastal systems, underscoring its potential to refine fine-scale hydrodynamics while ensuring consistency with larger-scale ocean processes (Petton et al., 2023).

The organization of these multi-model studies is identified by the coastal modelling community as a need. Firstly, to tackle common assessments of the wide range of overlapping (global/basin/regional and local) models that are available for users in some costal zones. Secondly, these multi-model validation exercises, comparing the performance of global/regional "core" model forecasts (i.e. from services such as the Copernicus Marine one) and coastal model solutions, nested into the formers, are useful to identify the potential added value (and the limitations) of performed coastal downscaling with respect to the "parent" core operational solutions, in which high-resolution coastal models are nested.

Frishfelds et al. (2025) highlight the benefits of on-demand coastal modeling employing two-way nesting, emphasizing its capacity to dynamically refine coastal processes while maintaining consistency with larger-scale ocean simulations. This

approach enhances the accuracy and reliability of high-resolution forecasting systems, facilitating improved representation of
fine-scale coastal dynamics.
In that sense, these multi-model intercomparison exercises are key elements for many initiatives, such as the Horizon Europe
Project FOCCUS (Forecasting and Observing the Open-to-Coastal Ocean for Copernicus Users, https://foccus-project.eu/)
Project, that have in their core the enhancing of existing coastal downscaling capabilities, developing innovative coastal
forecasting products based on a seamless numerical forecasting from regional models of the Copernicus Marine Service
covering the EU regional seas, to Member States coastal forecasting systems (authors can add here any other pertinent reference
from literature). Espino et al. (2022) emphasized the significance of extending Copernicus Marine Environmental Monitoring
Service (CMEMS) products to coastal regions, highlighting the integration of high-resolution models and observational data
to improve coastal forecasting capabilities. Their work underscores the importance of tailoring operational ocean models to
better capture nearshore dynamics, ensuring more accurate and actionable predictions for end-users. Furthermore, multi-model
studies focused on extreme event simulations provide valuable insights into the performance of operational forecasting
systems. For instance, Sotillo et al. (2021) examined the record-breaking Western Mediterranean Storm Gloria by evaluating
five different model systems, including Copernicus Marine Service products (global, regional Mediterranean, and Atlantic IBI
solutions) alongside two coastal nested models. Such studies play a crucial role in assessing model accuracy, leveraging local
HF radar observations, and informing future improvements to regional and coastal forecasting services.
Furthermore, and from an end-user perspective, multi-model study cases focused on extreme event simulations, such as the
one performed by Sotillo et al. (2021) focused on the record-breaking Western Mediterranean Storm Gloria, allow to identify
strengths and limitations of model solutions delivered by operational forecast services available in zones affected by extreme
events; for instance, in the referred study case, 5 model systems were considered (including systems both from the Copernicus
Marine service -with usages of the Global and the regional Mediterranean and Atlantic IBI solutions- and 2 coastal services
nested into the regional solutions). This kind of multi-model study cases certainly help to enhance product quality assessments
(in this Gloria Storm case, making extensive use of the local HF radar capabilities), increasing the knowledge about the model
systems in operations, and outlining future model service upgrades (both in the regional and coastal services) aimed at
achieving a better coastal forecasting of extreme events.
**3.4 Unstructured-Grid Models for Cross-Scale Coastal Dynamics**
The use of unstructured-grid models is crucial for cross-scale modelling and effectively addressing the interactions between
estuaries and the open ocean. One key aspect is the accurate representation of freshwater transformation from rivers, which is
often oversimplified in ocean models by specifying river runoff as a point source. Unstructured-grid models, while often
employing lower-order spatial discretizations due to interpolation complexities on irregular meshes, provide enhanced
flexibility in resolution placement and transition zones. This allows them to effectively capture subtidal, tidal, and intermittent
processes in coastal and estuarine environments, supporting a more realistic representation of estuarine dynamics and improved
coupling with estuarine models.

Compared to curvilinear and Cartesian grids, unstructured grids excel in resolving complex bathymetric features without significant grid stretching. Since bathymetry plays a fundamental role in governing the dynamics of estuaries and the near coastal zone, unstructured grid models offer greater accuracy and computational efficiency in numerical forecasting. Their flexibility also enables more effective resolution of multiscale dynamic features. Fine spatial resolution in unstructured-grid models allows for the resolution of secondary (transversal) circulation in estuaries and straits, thereby improving mixing and enhancing the representation of long-channel changes in stratification, as demonstrated by Haid et al. (2020). Zhang et al. (2016) have emphasized the role of cross-scale modeling in capturing multi-scale hydrodynamic interactions, particularly in tidal straits, where unstructured-grid models enhance the representation of exchange flows and stratification dynamics. As Ilicak et al. (2021) have shown, these advancements contribute to more precise simulations of estuarine and strait dynamics. Recent research has further elucidated the mechanisms governing secondary circulation in tidal inlets. Chen et al. (2023) demonstrated that subtidal secondary circulation can arise due to the covariance between eddy viscosity and velocity shear, even in predominantly well-mixed tidal environments. This finding highlights the necessity of incorporating high-resolution turbulence parameterizations within unstructured-grid models to accurately capture submesoscale and cross-channel processes, thereby improving the fidelity of numerical simulations in complex coastal and estuarine systems.

However, the construction of grids and ensuring reproducibility in unstructured-grid modelling still present challenges. Grid generation is not always fully automated, and subjective decisions are often made based on the specific research problem, applications, and intended services. The development of more objective grid construction methods and reproducibility standards is an ongoing concern in unstructured-grid modelling (Candy and Pietrzak, 2018). One significant advancement is the introduction of the JIGSAW mesh generator (Engwirda, 2017), which enables the creation of high-quality unstructured grids designed to satisfy specific numerical requirements. JIGSAW produces centroidal Voronoi tessellations with well-centred, orthogonal cell geometries that are particularly suitable for mimetic finite-volume schemes. JIGSAW incorporates mesh optimisation strategies tailored to geophysical fluid dynamics and has been increasingly adopted in ocean modelling applications.

The generation of unstructured meshes is a critical component in configuring coastal and estuarine ocean models, as it directly influences numerical accuracy, computational efficiency, and the ability to represent complex shoreline and bathymetric features. Tools such as OceanMesh2D offer MATLAB-based workflows for high-quality, two-dimensional unstructured mesh generation, facilitating user control over mesh density and coastal geometry resolution (Roberts et al., 2019). Similarly, OPENCoastS provides an open-access, automated service that streamlines the setup of coastal forecast systems, integrating mesh generation, model configuration, and forecast production (Oliveira et al., 2019, 2021). The OCSMesh software developed by NOAA represents another important advancement. It enables data-driven, automated unstructured mesh generation tailored for coastal ocean modeling, offering a robust framework to ensure mesh quality, reproducibility, and interoperability with NOAA modeling systems (Mani et al., 2021). Together, these developments represent the ongoing progress toward objective, reproducible, and user-oriented mesh generation in support of high-resolution coastal ocean modelling.

**3.5 Observing System Simulation Experiments, Observing System Experiments and Data Assimilation**

Data assimilation in coastal regions presents challenges due to the presence of multiple scales and competing forcings from open boundaries, rivers, and the atmosphere, which are often imperfectly known (Moore and Martin, 2019). Data assimilation is particularly challenging in tidal environments (especially for meso- and macro-tidal environments; and not so in micro-tidal coastal zones (De Mey et al., 2017, Stanev et al, 201, Holt et al, 2012). Studies by Oke et al. (2002), Wilkin et al. (2005), Shulman and Paduan (2009), Stanev et al. (2015, 2016), and others have demonstrated the value of assimilating HF radar observations to improve the estimation of the coastal ocean state.

Observing System Simulation Experiments (OSSE) and Observing System Experiments (OSE) are widely used techniques for assessing and optimizing ocean observational systems. OSSEs involve numerical simulations that test the potential impact of hypothetical observations on forecast models before actual observations are made, enabling improved planning and cost-effective observational strategies. In contrast, OSEs assess the impact of existing observations by systematically removing certain datasets from assimilation systems and evaluating the resulting degradation in model performance. OSSE and OSE have the capability to incorporate diverse observing systems, including satellite-based observations, HF radars, buoys with low-cost sensors, autonomous vehicles, and more. These approaches provide valuable insights for refining data assimilation techniques and guiding the development of future observational networks. For further details, we refer readers to Oke and Sakov (2012) and Fujii et al. (2019), who provide comprehensive discussions on the methodologies and applications of OSSEs and OSEs in operational oceanography. an in-depth review of OSSE methodologies and insights into how OSSE and OSE methodologies contribute to improving ocean forecasting, designing observational systems, and refining numerical models is given in Zeng et al (2020). These approaches can help identify gaps in existing coastal observing networks, assess operational failure scenarios, and evaluate the potential of future observation types. Pein et al. (2016) used an OSE-type approach to investigate the impact of salinity measurements in the Ems Estuary on the reconstruction of the salinity field, identifying observation locations that are more suitable for model-data synthesis. This type of analysis can contribute to the design and optimization of both existing and future observational arrays, especially in coastal regions where fine resolution is required.

**3.5 Riverine forcing and its role in coastal ocean Modeling**

Rivers play a critical role in shaping coastal circulation and stratification by delivering freshwater, nutrients, and sediments that influence estuarine and shelf dynamics. The treatment of riverine inputs in ocean models remains a key source of uncertainty, especially when estuarine plume dynamics and mixing processes are unresolved. In many coarse-resolution systems, river discharge is prescribed via simplified surface or salinity fluxes, which may misrepresent the spatial structure and strength of river plumes (Sun et al., 2017; Verri et al., 2020). To address this, high-resolution and regional-scale models increasingly incorporate momentum-carrying river inflows or artificial estuarine channels (Herzfeld, 2015; Sobrinho et al., 2021). For instance, Nguyen et al. (2024) demonstrated how high-resolution modeling in the German Bight captures the hydrodynamic and biogeochemical responses to extreme river discharge events, showing significant implications for salinity,

stratification, and nutrient dispersion during floods. These findings underscore the importance of resolving riverine inflow variability and extreme events in coastal ocean prediction systems.

Recent work has also focused on operational strategies for river forcing (Matte et al., 2024), including real-time discharge data integration (e.g., from GloFAS; Harrigan et al., 2020), and estuary box models that approximate sub-grid plume behavior (Sun et al., 2017). These approaches aim to enhance predictive capabilities while maintaining computational feasibility in global-to-coastal modeling chains. Choosing the appropriate river input strategy is therefore application-dependent and strongly influenced by spatial resolution and target phenomena.

**3.6 Integration of AI in Coastal Modeling and Forecasting**

The integration of artificial intelligence (AI) and machine learning (ML) techniques in ocean and coastal forecasting has rapidly evolved, providing novel methodologies for improving predictive accuracy, computational efficiency, and data assimilation in operational models. Recent advances in AI-based approaches for parameterizing subgrid-scale processes, hybrid modelling techniques, and ensemble forecasting highlight the transformative potential of these methods in coastal modelling (Heimbach et al., 2024).

Machine learning applications in coastal ocean modeling primarily focus on two domains: (1) enhancing conventional physical models by integrating ML-based parameterizations and error corrections, and (2) fully data-driven approaches that employ neural networks as surrogate models (Zanna & Bolton, 2020; Bolton & Zanna, 2019). The former leverages ML techniques to optimize numerical model performance by improving subgrid parameterizations, bias correction, and data assimilation strategies, while the latter explores the potential of deep learning algorithms such as Fourier Neural Operators (FNOs) and Transformer-based architectures for high-resolution ocean forecasting (Bire et al., 2023; Wang et al., 2024).

Data assimilation, a critical component of operational forecasting, benefits from AI-enhanced methodologies that improve state estimation and predictive skill. AI-driven data assimilation frameworks, such as the combination of deep learning with variational assimilation (4D-VarNet) (Fablet et al., 2022), have demonstrated superior performance in coastal and regional models. Hybrid approaches incorporating AI techniques into numerical models have been applied to refine coastal simulations, allowing for better representation of multi-scale interactions (Brajard et al., 2021). Furthermore, convolutional neural networks (CNNs) have been successfully used for downscaling sea surface height and currents in coastal areas, addressing challenges related to observational gaps and improving model resolution (Yuan et al., 2024).

Coastal high-resolution models often suffer from errors stemming from inaccuracies in numerics, forcing (e.g., open boundaries, meteorological inputs), and unresolved physical processes. AI-based methods have been increasingly applied to address these challenges, particularly in the realm of subgrid-scale parameterization. AI-enabled parameterizations of mesoscale and submesoscale processes using deep learning techniques, such as residual networks and generative adversarial networks (GANs), have shown promising results in reducing bias in numerical simulations (Gregory et al., 2023; Brajard et al., 2021). Additionally, hybrid methods combining physics-based models with ML correction schemes have demonstrated improved predictive skill for regional and coastal ocean models (Perezhogin et al., 2023).

The use of ML for extreme event prediction has gained increasing attention in the context of operational coastal forecasting. AI models trained on historical storm data and high-resolution numerical simulations have been utilized to enhance storm surge predictions and improve early warning systems (Xie et al., 2023). Transformer-based models, originally developed for atmospheric forecasting, have been adapted for ocean applications, achieving competitive skill in eddy-resolving ocean simulations (Wang et al., 2024).

The integration of AI in ensemble forecasting further contributes to uncertainty quantification, providing probabilistic predictions for extreme coastal events. Bayesian inference techniques, combined with ML-based ensemble prediction, offer a framework for optimizing multi-model ensembles and reducing systematic errors in operational forecasts (Bouallègue et al., 2024; Penny et al., 2022). The synergy between ML-driven emulators and traditional ensemble forecasting techniques has the potential to enhance coastal hazard predictions, particularly in regions prone to high-impact events.

Despite the advancements in AI for coastal modeling, several challenges remain. The interpretability and robustness of ML-based solutions need further improvement, particularly for operational applications requiring high levels of reliability (Bonavita, 2023). Additionally, integrating ML models with real-time observational data streams, including remote sensing and high-frequency radar (HFR) networks, remains an ongoing area of research (Reichstein et al., 2019). The extension of ML-based ocean forecasting to seasonal and interannual time scales also poses challenges related to long-term stability and physical consistency (Beucler et al., 2024).

## 4 Summary and outlook

The critical importance of high-resolution coastal modelling is demonstrated in addressing the complexities of dynamic coastal systems. Coastal areas are shaped by the interplay of mesoscale and submesoscale processes, strong tidal currents, atmospheric and hydrologic forcing, and significant anthropogenic pressures. Advanced techniques, including turbulence closure models for capturing vertical mixing and parameterizations of bottom roughness and vegetation drag for representing energy dissipation, are essential for accurately modelling these systems. The nonlinear interactions between tidal currents and wind waves emerge as a particularly influential factor, affecting ocean circulation and improving the accuracy of sea surface temperature predictions.

It is shown that the integration of high-resolution observational data, such as HF radar for surface currents and the SWOT satellite mission for sea surface topography, has the potential of substantially enhancing the resolution and reliability of coastal models. These data facilitate a detailed characterization of processes in transition zones spanning estuaries, nearshore areas, and the open ocean. Improved coupling between regional and local models has advanced the representation of boundary conditions and enabled simulations of small-scale dynamics, essential for capturing the complexity of the coastal continuum. The application of data assimilation techniques addresses the rapid variability inherent in coastal processes, highlighting the challenges and limitations of predictability in these highly dynamic environments. Strategies to extend the accuracy of short-term and localized forecasts are provided, leveraging multiscale data integration to refine predictions. The ability to simulate interactions between atmospheric conditions, hydrological inputs, and oceanographic processes strengthens the foundation for

more accurate modelling. This contribution underscores the importance of bridging observational and modelling gaps to achieve a comprehensive understanding of coastal systems. It highlights the necessity of integrating small-scale dynamics with broader processes to better inform sustainable coastal management practices. By aligning advanced techniques with high-resolution data, this work offers a pathway for more robust representations of coastal ocean dynamics and supports informed decision-making in the face of growing environmental and societal challenges.

Several directions for advancing coastal ocean modelling to address evolving environmental and societal challenges are highlighted. Future efforts should focus on integrating emerging observational technologies, such as high-resolution satellites (e.g., SWOT), autonomous platforms like gliders and drones, and hyperspectral imaging. These tools, combined with machine learning techniques for data analysis, can bridge gaps in spatial and temporal data coverage, providing a richer understanding of coastal dynamics.

Developing coupled modelling systems that seamlessly integrate atmospheric, hydrological, and oceanographic processes will be essential for capturing the complexities of the land-ocean continuum. Incorporating river runoff, estuarine dynamics, and nearshore processes into such systems will significantly enhance the scope and accuracy of predictions. Addressing computational challenges associated with high-resolution modelling is equally critical; this includes leveraging high-performance computing, cloud-based processing, and optimizing numerical schemes to achieve efficient and precise simulations.

Improving data assimilation techniques through ensemble approaches and probabilistic forecasting is another priority. These methods will better integrate multiscale observational data, reduce uncertainties, and enhance the reliability of predictions in dynamic environments. Concurrently, there is a pressing need to explore the impacts of climate change on coastal systems, including sea-level rise, increased storm intensity, and shifting precipitation patterns. Understanding these impacts will guide the development of adaptive strategies and strengthen resilience in vulnerable coastal zones.

The future of coastal modelling also depends on fostering interdisciplinary collaboration, engaging expertise from oceanography, meteorology, hydrology, and ecology. By aligning scientific research with societal needs and practical applications, collaborative frameworks can ensure the relevance and effectiveness of modelling efforts. Additionally, applying artificial intelligence to optimize model parameterization, grid design, and predictive analyses will unlock new capabilities for simulating small-scale processes like sediment transport and ecosystem responses.

Finally, enhancing global and regional coordination for coastal monitoring and modelling will be vital. Strengthening networks to ensure consistency in data and modelling approaches can foster international collaboration, facilitating the exchange of best practices and resources. These collective advancements promise to deepen our understanding of coastal systems and provide robust tools to manage and protect these critical areas sustainably in the face of ongoing and future challenges.

489
490

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

**Competing interests**

The contact author has declared that none of the authors has any competing interests.

**Author contributions**

JS conceptualized the study, analyzed data, and wrote this article. All authors contributed to the writing of the article and quality control.

**Competing interests**

The contact author has declared that neither of the authors has any competing interests.

**Disclaimer**

Views and opinions expressed are however those of the author(s) only and do not necessarily reflect those of the European
Union or the European Health and Digital Executive Agency (HaDEA). Neither the European Union nor HaDEA can be held
responsible for them.

Publisher's note: Copernicus Publications remains neutral with regard to jurisdictional claims made in the text, published
maps, institutional affiliations, or any other geographical representation in this paper. While Copernicus Publications makes
every effort to include appropriate place names, the final responsibility lies with the authors.

Acknowledgements
JS and AM acknowledge Horizon Europe Project FOCCUS "Forecasting and observing the open-to-coastal ocean for
Copernicus users" (Grant Agreement 101133911).