# Peer review of "Solving Coastal Dynamics: Introduction to High Resolution Ocean"

_State of the Planet, 2024_

## Referee Comment (RC2)

The paper aims to introduce high resolution ocean models used in operational coastal forecasting applications, describing the spatial scales, and key processes addressed by high-resolution coastal and regional models.

Some numerical modeling techniques essential for capturing such small-scale coastal dynamics (including parameterization of turbulence and mixing effects, the modelling of tidal dynamics or the accounting of variable freshwater river contributions) are highlighted. Being also discussed the role of some monitoring tools (especially ADCPs, HF Radar, or the coming SWOT coastal altimetry), that linked to data assimilation techniques and through OSEs and OSSEs can help to improve prediction accuracy in the coastal zone.

A list of ocean models commonly used for coastal and regional modelling is provided, together with some insights into the fine resolution nesting techniques and the use of unstructured-grid models to simulate the coastal zone.

By addressing these topics, the paper effectively achieves its main goal, introducing the status of high-resolution ocean forecasting services aimed to solve coastal dynamics, as mentioned in the paper title.

Some specific points that can be addressed in more detail by the authors in the updated manuscript:

- Links with emerging AI-based solutions. In the abstract it is said: "This work underscores the potential of advancing coastal forecasting systems through interdisciplinary innovation, paving the way for enhanced scientific understanding and practical applications" and in the conclusion is stated: "applying artificial intelligence to optimize model parameterization, grid design, and predictive analyses will unlock new capabilities for simulating small-scale processes like sediment transport and ecosystem responses." However, in the paper sections, there is no reference to any AI application. Can the authors include in the manuscript some information about the links between the new AI-based solutions and coastal modelling? The inclusion of some information on AI applications (and references to related literature or to on-going initiatives) will enhance the proposed review, and will certainly increase its interest to more potential readers.

- **Section 3.3. Fine resolution nested models, downscaling and upscaling.**
  This section, on finer resolution nested models, downscaling and upscaling techniques, would need some refinement.

After the initial paragraph explaining coastal downscaling through one-way nesting, the authors describe an example of increasing resolution in a regional system (not clearly referred; see next point). Then a paragraph on 2-way nesting coupling, describing an example (not sufficiently referred), and a final paragraph on the AGRIF tool from NEMO.

I see ok the proposed section organization, but I would ask the authors to keep the same level of review detail(/references) of the first paragraph along the whole section. For instance, some more examples can be included/referred for the 2-way nesting part.

I would also ask the authors to add in this section some information about multi-model intercomparison exercises (this section downscaling/upscaling may be the place to mention this working line that combines the use of global, basin, regional and coastal models).

Some pieces of information about multi-model intercomparison exercises that I would recommend the authors to include in this section can be in the line:

> The organization of these multi-model studies is identified by the coastal modelling community as a need. Firstly, to tackle common assessments of the wide range of overlapping (global/basin/regional and local) models that are available for users in some costal zones. Secondly, these multi-model validation exercises, comparing the performance of global/regional "core" model forecasts (i.e. from services such as the Copernicus Marine one) and coastal model solutions, nested into the formers, are useful to identify the potential added value (and the limitations) of performed coastal downscaling with respect to the "parent" core operational solutions, in which high-resolution coastal models are nested.

> In that sense, these multi-model intercomparison exercises are key elements for many initiatives, such as the **HE FOCCUS** (Forecasting and Observing the Open-to-Coastal Ocean for Copernicus Users) Project, that have in their core the enhancing of existing coastal downscaling capabilities, developing innovative coastal forecasting products based on a seamless numerical forecasting from regional models of the Copernicus Marine Service covering the EU regional seas, to Member States coastal forecasting systems (authors can add here any other pertinent reference from literature).

> Furthermore, and from an end-user perspective, multi-model study cases focused on extreme event simulations, such as the one performed by **Sotillo et al. (2021)** focused on the record-breaking Western Mediterranean Storm Gloria, allow to identify strengths and limitations of model solutions delivered by operational forecast services available in zones affected by extreme events; for instance, in the referred study case, 5 model systems were considered (including systems both from the Copernicus Marine

service -with usages of the Global and the regional Mediterranean and Atlantic IBI solutions- and 2 coastal services nested into the regional solutions). This kind of multi-model study cases certainly help to enhance product quality assessments (in this Gloria Storm case, making extensive use of the local HF radar capabilities), increasing the knowledge about the model systems in operations, and outlining future model service upgrades (both in the regional and coastal services) aimed at achieving a better coastal forecasting of extreme events.

Considering the present organization of section 3 in the manuscript, I would suggest the authors adding this reflection about multi-model studies after Line 229 statement. Anyway, please, feel free to elaborate on it, including more references to different multi-model intercomparison exercises (there are several examples in the literature for different zones).

Sotillo MG, Mourre B, Mestres M, Lorente P, Aznar R, García-León M, Liste M, Santana A, Espino M and Álvarez E (2021) Evaluation of the Operational CMEMS and Coastal Downstream Ocean Forecasting Services During the Storm Gloria (January 2020). *Front. Mar. Sci.* 8:644525. doi: 10.3389/fmars.2021.644525

- **ln 231.** Enhancing the horizontal resolution of the North Sea operational model from 7 to 1.5 kilometers has shown improvements in off-shelf regions, but biases persist over the shelf area, indicating the need for further enhancements in surface forcing, vertical mixing, and light attenuation.
  Here when saying "North Sea operational model" are the authors referring to the Copernicus Marine NWS-MFC forecasting model system? If so, and the increase of resolution mentioned is the one documented in **Tonani et. al. (2019),** please, refer properly to such work (reference below). If not, please specify which system and resolution increase is here being mentioned.
  Tonani, M., Sykes, P., King, R. R., McConnell, N., Péquignet, A.-C., O'Dea, E., Graham, J. A., Polton, J., and Siddorn, J.: The impact of a new high-resolution ocean model on the Met Office North-West European Shelf forecasting system, Ocean Sci., 15, 1133–1158, https://doi.org/10.5194/os-15-1133-2019, 2019.

- **ln 257.** Fine spatial resolution in unstructured-grid models allows for the resolution of secondary (transversal) circulation in estuaries and straits **(Ilicak et al. 2021),** thereby improving mixing and enhancing the representation of long-channel changes in stratification, as demonstrated by Haid et al.
  The Ilicak et al. 2021 paper nicely illustrates how a high-resolution unstructured grid model is used to enhance the simulation of circulation across the Turkish Strait System that communicate both Mediterranean and Black Seas.

Ilicak, M., Federico, I., Barletta, I., Mutlu, S., Karan, H., Ciliberti, S. A., Clementi, E., Coppini, G., & Pinardi, N. (2021). Modeling of the Turkish Strait System Using a High Resolution Unstructured Grid Ocean Circulation Model. *Journal of Marine Science and Engineering*, 9(7), 769. https://doi.org/10.3390/jmse9070769

- **In 266.** Data assimilation in coastal regions presents challenges due to the presence of multiple scales and competing forcings from open boundaries, rivers, and the atmosphere, which are often imperfectly known (Moore and Martin, 2019).
  In this point, I would suggest adding specific mention to tides as one of the main challenges for data assimilation. There are many references in the literature to the (absence of) data assimilation in tidal coastal zones. One statement like the following one can be added to the manuscript: **Data assimilation is particularly challenging in tidal environments (especially for meso- and macro-tidal environments; and not so in micro-tidal coastal zones).** - included selected update references...-

- Some parts of the manuscript lack citation. This is especially so, for instance in section 2. Unlikely, some other sections of the manuscript provide much more level of references to previous works than Section 2. Below, some options for balancing this different level of citation, including some pertinent reference to address the following points can be:

**In 82.** High-resolution services in the coastal ocean operate at various spatial scales depending on the specific applications and objectives (**Sotillo, 2022**).

This Special Issue (Sotillo, 2022) entitled "ocean modelling in support of operational ocean and coastal services", compiles 11 recent papers on operational coastal services based on high-resolution models. Its citation here can certainly provide readers with insights about scales, objectives and applications, as stated in this sentence with no reference.

Sotillo, M. G. (2022). Ocean Modelling in Support of Operational Ocean and Coastal Services. *Journal of Marine Science and Engineering*, *10*(10), 1482. https://doi.org/10.3390/jmse10101482

**In 135.** Accounting for high-resolution atmospheric forcing into coastal models is essential for accurately capturing local meteorological dynamics, including wind patterns, temperature gradients, and precipitation rates. Such detailed atmospheric data drive fundamental processes like heat and momentum fluxes

**(García-León et al. 2022),** profoundly influencing coastal hydrodynamics, sediment transport, and ecosystem response.

The proposed García-León et al. work demonstrates that the usage of a new high-resolution atmospheric forcing, together with the update of bulk formulae to compute surface fluxes, have positive impacts across different high-resolution model systems for ports.

García-León, M., Sotillo, M. G., Mestres, M., Espino, M., & Fanjul, E. Á. (2022). Improving Operational Ocean Models for the Spanish Port Authorities: Assessment of the SAMOA Coastal Forecasting Service Upgrades. *Journal of Marine Science and Engineering*, *10*(2), 149. https://doi.org/10.3390/jmse10020149.

This is one example, among many others in the literature (authors should fell free to add more), that can support such statement on atmospheric forcing. I would recommend the authors look for some more references to complete this aspect related to the improvement of the atmospheric forcing in coastal high-resolution modeling. Indeed, it would be interesting if in a review paper like this, it is mentioned that atmospheric forcing can be seen today unfortunately as a common limitation for high-resolution coastal modelling. Especially, when coastal modelers are aiming and working on (as stated at the end of this section 2; in 158):

"Simulations at grid resolutions that would sufficiently resolve the coastal sub mesoscale would require horizontal grid resolutions of approximately 1-10 meters in estuaries and 0.1-1 kilometer in coastal shelf domains. However, achieving such high resolutions poses significant computational challenges and resource demands".

But coastal modelers typically can rely only on atmospheric forcing data from national/regional operational services, which have resolutions of around 2-5 km resolution, often being this the best available option (or even lower resolution data when no other alternatives are available). Authors may also link this point on atmospheric forcing limitations with on-going AI initiatives to improve coastal winds.

**In 149.** The interactions between tidal forcing, river flow and estuarine geometry result in intricate and variable periodic patterns (as shown in **Campuzano et al. 2022 for the Western Iberian Buoyant Plume and in Sotillo et al. 2021 for the whole European Atlantic façade**).

Campuzano et al. and Sotillo et al. works (on the simulation of the Western Iberian Buoyant Plume formed by the contribution of several rivers, and the sensitivity of IBI model to different river forcing data) can provide some illustration of the intricate

and variable patterns resulting between river flows, estuarine geometries; and all in regions with marked tidal influence.

Campuzano, F., Santos, F., Simionesei, L., Oliveira, A. R., Olmedo, E., Turiel, A., Fernandes, R., Brito, D., Alba, M., Novellino, A., & Neves, R. (2022). Framework for Improving Land Boundary Conditions in Ocean Regional Products. *Journal of Marine Science and Engineering*, *10*(7), 852. https://doi.org/10.3390/jmse10070852

Sotillo, M. G., Campuzano, F., Guihou, K., Lorente, P., Olmedo, E., Matulka, A., Santos, F., Amo-Baladrón, M. A., & Novellino, A. (2021). River Freshwater Contribution in Operational Ocean Models along the European Atlantic Façade: Impact of a New River Discharge Forcing Data on the CMEMS IBI Regional Model Solution. *Journal of Marine Science and Engineering*, *9*(4), 401. https://doi.org/10.3390/jmse9040401

- Some other references that may enhance the proposed introductory/review scope of the paper may be the following ones:

**In 197**. The use of HFR networks has become an essential element of coastal ocean observing systems, contributing to high-level coastal services (Stanev et al., 2016; Rubio et al., 2017; **Reyes et al., 2022**)

I would suggest including this more recent review work of Reyes et al. on existing HFR data multidisciplinary science-based applications in the Mediterranean Sea, primarily focused on meeting end-user and science-driven requirements, addressing regional challenges in maritime safety, extreme hazards and environmental transport processes.

Reyes, E., Aguiar, E., Bendoni, M., Berta, M., Brandini, C., Cáceres-Euse, A., Capodici, F., Cardin, V., Cianelli, D., Ciraolo, G., Corgnati, L., Dadić, V., Doronzo, B., Drago, A., Dumas, D., Falco, P., Fattorini, M., Fernandes, M. J., Gauci, A., Gómez, R., Griffa, A., Guérin, C.-A., Hernández-Carrasco, I., Hernández-Lasheras, J., Ličer, M., Lorente, P., Magaldi, M. G., Mantovani, C., Mihanović, H., Molcard, A., Mourre, B., Révelard, A., Reyes-Suárez, C., Saviano, S., Sciascia, R., Taddei, S., Tintoré, J., Toledo, Y., Uttieri, M., Vilibić, I., Zambianchi, E., and Orfila, A.: Coastal high-frequency radars in the Mediterranean – Part 2: Applications in support of science priorities and societal needs, Ocean Sci., 18, 797–837, https://doi.org/10.5194/os-18-797-2022, 2022.

- Minor (typo) Points.

In 15 typo: "introduce key"

In 17 typo: "for the"

---

## Author Comment (AC1)

**Authors' response to Reviewer #1**

We sincerely appreciate the time and effort that Reviewer #1 has taken to evaluate our manuscript, "Solving Coastal Dynamics: Introduction to High Resolution Ocean Forecasting Services". The constructive comments provided have significantly contributed to improving the clarity, accuracy, and overall quality of our manuscript. Below, we provide a detailed point-by-point response to each of the reviewer's comments, along with explanations of the corresponding revisions made in the manuscript.

- RC1: 'Comment on sp-2024-44', Anonymous Referee #1, 13 Jan 2025  reply

The paper highlights the requirements that coastal models need to meet in order to properly represent complex phenomena encountered in the coastal ocean. The approach to coastal ocean modelling differs from open ocean in many aspects and the paper discusses modelling strategies for the coastal ocean. The presented work aims to support the development of more robust and adaptable tools for coastal forecasting.

Authors' response: We acknowledge and appreciate the reviewer's insightful feedback and have addressed each point carefully. Our primary focus in the revisions has been to enhance clarity, ensure scientific accuracy, and correct minor typographical errors.

This aim is achieved, in my opinion, and the paper reads very well and is well structured. There are some minor deficiencies that need to be addressed before the publication and I list them below.

Ln 15 introduceskey >> introduces key

Authors' response: This correction has been made in the revised manuscript.

Ln 17 forthe >> for the

Authors' response: The requested correction has been implemented in the manuscript.

Ln 39 I am not sure if the terms 'anthropogenic pressures' and 'natural drivers' are used here in the right context. I suggest these are replaced with direct and indirect anthropogenic impacts, respectively, which I think was the intended meaning by the authors. Some of the processes listed do have their own natural variability too, so the authors can, of course, can reflect it in the revised text

Authors' response: We have made the suggested revision in the text.

Ln 52 add 'the' before 'models'

Authors' response:  have made the suggested revision in the text.

Ln 57 Again, the use of the term 'natural changes' in this paragraph should be revised in line with the comment above

Authors' response:  We have made the suggested revision in the text.

Ln 63 use capital letters for MSFD

Authors' response: We have made the suggested revision in the text.

Ln 91 one of my main comments is related with the statement included in this paragraph. The authors say that the resolution of the coastal scale models typically range from a few to tens of km. This is way too coarse. In line 42 the authors state, rightly so, that coastal scale models need to resolve submesoscale processes, i.e. the processes of the scale <100 km, or perhaps even 1 to 10 km, as the authors state in line 100. In order to capture these, the resolution of the numerical models has to be at least 10 times greater, e.g. a 100 km scale requires the model of <10 km resolution, a 10 km the model of < 1 km resolution, etc. Coastal models certainly cannot be of the tens of km resolution. Computing power increases all the time. Developing a coastal model of the resolution of tens of km is simply bad practice. To put it in the perspective, the Copernicus global model is < 10km and regional models are < 5km. Coastal models should typically be c. 1km and less.

Authors' response: We have made the suggested revision in the text.

Ln 135 into >> in the

Authors' response: This correction has been applied as suggested.

Ln 159-160 Here the authors correctly state the required horizontal resolutions in contradiction to the statements discussed above

Authors' response: Thank you. Now this is consistent.

Section 3.2 At least two important omissions in Table 1, NEMO and POM models

Authors' response: This correction has been applied as suggested.

Ln 231 which operational North Sea model?

Ln 240 references?

Ln 242 'developed ..' >> 'designed to exchange'

Authors' response: This correction has been applied as suggested.

Ln 269 an explanation is needed on what OSSE and OSE are for the readers that are non-familiar. Especially that the paper is addressed to the readers less familiar with ocean modelling since most expert modellers would be well aware of the issues addressed in this paper. At the least, references should be added to the publications or online resources that introduce the concepts of OSSE and OSE.

Authors' response: We appreciate this valuable suggestion and agree that providing a clear explanation of OSSE (Observing System Simulation Experiments) and OSE (Observing System Experiments) is essential for readers who are less familiar with ocean

modelling. In the revised manuscript, we have added explanatory sentences to clarify these concepts, and additional references, supporting this.

Replace 'modeling' with 'modelling' across the manuscript

Authors' response: This correction has been applied as suggested.

---

## Author Comment (AC2)

**Authors' Response to Reviewer #2**

We sincerely appreciate the constructive feedback provided by Reviewer #2. The comments have contributed to improving the clarity, depth, and comprehensiveness of our manuscript. We have carefully considered each point and made the necessary revisions to enhance the scientific rigor of the paper. In particular, we have expanded the discussion on the limitations of atmospheric forcing in high-resolution coastal modeling, incorporated recent advances in numerical and statistical methods for improving coastal wind fields, and strengthened the section on multi-model intercomparison exercises. Additional references have been integrated to provide a more comprehensive overview of relevant developments in the field. Minor typographical errors have also been corrected.

Note: The reviewer's comments are presented in black, and our responses are provided in blue. Below, we address each point systematically.

The paper aims to introduce high resolution ocean models used in operational coastal forecasting applications, describing the spatial scales, and key processes addressed by high-resolution coastal and regional models.

Some numerical modeling techniques essential for capturing such small-scale coastal dynamics (including parameterization of turbulence and mixing effects, the modelling of tidal dynamics or the accounting of variable freshwater river contributions) are highlighted. Being also discussed the role of some monitoring tools (especially ADCPs, HF Radar, or the coming SWOT coastal altimetry), that linked to data assimilation techniques and through OSEs and OSSEs can help to improve prediction accuracy in the coastal zone.

A list of ocean models commonly used for coastal and regional modelling is provided, together with some insights into the fine resolution nesting techniques and the use of unstructured-grid models to simulate the coastal zone.

By addressing these topics, the paper effectively achieves its main goal, introducing the status of high-resolution ocean forecasting services aimed to solve coastal dynamics, as mentioned in the paper title.

Authors response: We acknowledge and appreciate the reviewer's insightful feedback and have addressed each point carefully. Our primary focus in the revisions has been to enhance clarity, ensure scientific accuracy, and correct minor typographical errors.

Some specific points that can be addressed in more detail by the authors in the updated manuscript:

- Links with emerging AI-based solutions. In the abstract it is said: "This work underscores the potential of advancing coastal forecasting systems through interdisciplinary innovation, paving the way for enhanced scientific understanding and practical applications" and in the conclusion is stated: "applying artificial intelligence to optimize model parameterization, grid design, and predictive analyses will unlock new capabilities for simulating small-scale processes like sediment transport and ecosystem responses." However, in the paper sections, there is no reference to any AI application. Can the authors include in the manuscript some information about the links between the new AI-based solutions and coastal modelling? The inclusion of some information on AI applications (and references to related literature or to on-going initiatives) will enhance the proposed review, and will certainly increase its interest to more potential readers.

Authors response: We acknowledge the reviewer's suggestion and have addressed this point by introducing a dedicated section 3.6 on the integration of AI-based solutions in coastal modeling and forecasting. This new section outlines recent advancements in AI-driven approaches, including data assimilation improvements, hybrid modeling techniques, ensemble forecasting enhancements, and digital twins for coastal applications. Additionally, we have incorporated references to key studies demonstrating AI applications in ocean forecasting, such as the use of deep learning for subgrid parameterization (Bolton & Zanna, 2019), AI-based data assimilation frameworks (Brajard et al., 2021), and CNN-driven downscaling techniques for sea surface height and currents (Yuan et al., 2024). Furthermore, we discuss the role of machine learning in extreme event prediction, uncertainty quantification, and the optimization of high-resolution coastal models (Heimbach et al., 2024; Bire et al., 2023). These additions provide a more detailed perspective on the intersection of AI and coastal modeling, strengthening the manuscript's discussion on emerging methodologies and their implications for advancing predictive capabilities in coastal oceanography.

- **Section 3.3. Fine resolution nested models, downscaling and upscaling.**

This section, on finer resolution nested models, downscaling and upscaling techniques, would need some refinement.

After the initial paragraph explaining coastal downscaling through one-way nesting, the authors describe an example of increasing resolution in a regional system (not clearly referred; see next point). Then a paragraph on 2-way nesting coupling, describing an example (not sufficiently referred), and a final paragraph on the AGRIF tool from NEMO. I see ok the proposed section organization, but I would ask the authors to keep the same level of review detail(/references) of the first paragraph along the whole section. For instance, some more examples can be included/referred for the 2-way nesting part.

Authors response: We appreciate the reviewer's comments and have refined the section to ensure consistency in detail and references. The example of increasing resolution in a regional system has been clarified with explicit reference to the study in question: *For two-way nesting, we have expanded the discussion by including additional references to demonstrate its role in improving multi-scale interactions and model consistency (Herzfeld & Rizwi, 2019; Jeon et al., 2019). An additional example has been incorporated to highlight its application in coastal hydrodynamics and environmental indicator estimation (Petton et al., 2023).* The AGRIF tool in NEMO has been further detailed to align with the review depth of earlier sections, emphasizing its capabilities and supported by additional references.

I would also ask the authors to add in this section some information about multi-model intercomparison exercises (this section downscaling/upscaling may be the place to mention this working line that combines the use of global, basin, regional and coastal models).

Some pieces of information about multi-model intercomparison exercises that I would recommend the authors to include in this section can be in the line:

The organization of these multi-model studies is identified by the coastal modelling community as a need. Firstly, to tackle common assessments of the wide range of overlapping (global/basin/regional and local) models that are available for users in some costal zones. Secondly, these multi-model validation exercises, comparing the performance of global/regional "core" model forecasts (i.e. from services such as the Copernicus Marine one) and coastal model solutions, nested into the formers, are useful to identify the potential added value (and the limitations) of performed coastal downscaling with respect to the "parent" core operational solutions, in which high-resolution coastal models are nested.

In that sense, these multi-model intercomparison exercises are key elements for many initiatives, such as the **HE FOCCUS** (Forecasting and Observing the Open-to-Coastal Ocean for Copernicus Users) Project, that have in their core the enhancing of existing coastal downscaling capabilities, developing innovative coastal forecasting products based on a seamless numerical forecasting from regional models of the Copernicus Marine Service covering the EU regional seas, to Member States coastal forecasting systems (authors can add here any other pertinent reference from literature).

Furthermore, and from an end-user perspective, multi-model study cases focused on extreme event simulations, such as the one performed by Sotillo et al. (2021) focused on the record-breaking Western Mediterranean Storm Gloria, allow to identify strengths and limitations of model solutions delivered by operational forecast services available in zones affected by extreme events; for instance, in the referred study case, 5 model systems were considered (including systems both from the Copernicus Marine service - with usages of the Global and the regional Mediterranean and Atlantic IBI solutions- and

2 coastal services nested into the regional solutions). This kind of multi-model study cases certainly help to enhance product quality assessments (in this Gloria Storm case, making extensive use of the local HF radar capabilities), increasing the knowledge about the model systems in operations, and outlining future model service upgrades (both in the regional and coastal services) aimed at achieving a better coastal forecasting of extreme events.

Authors response: We acknowledge the importance of multi-model intercomparison exercises and have incorporated relevant information into the section.

A discussion has been added on the role of these exercises in assessing overlapping model systems across global, basin, regional, and coastal scales. Their utility in validating downscaling techniques and identifying the added value of high-resolution coastal models compared to parent systems has been highlighted. We have also referenced key initiatives, such as the HE FOCCUS project, which aim to enhance coastal forecasting by integrating seamless numerical modeling approaches. Additionally, we have included an example of multi-model assessments in extreme event simulations, such as the study on Storm Gloria (Sotillo et al., 2021), to illustrate their relevance in operational forecasting improvements.

Considering the present organization of section 3 in the manuscript, I would suggest the authors adding this reflection about multi-model studies after Line 229 statement. Anyway, please, feel free to elaborate on it, including more references to different multi-model intercomparison exercises (there are several examples in the literature for different zones).

Sotillo MG, Mourre B, Mestres M, Lorente P, Aznar R, García-León M, Liste M, Santana A, Espino M and Álvarez E (2021) Evaluation of the Operational CMEMS and Coastal Downstream Ocean Forecasting Services During the Storm Gloria (January 2020). *Front. Mar. Sci.* 8:644525. doi: 10.3389/fmars.2021.644525

Authors response:  We have incorporated the discussion on multi-model intercomparison studies after Line 229, as suggested. This addition highlights their relevance in evaluating model performance across different scales and their role in improving coastal forecasting capabilities. Additional references have been considered to provide a broader perspective on multi-model assessments in various regions.

- **In 231.** Enhancing the horizontal resolution of the North Sea operational model from 7 to 1.5 kilometers has shown improvements in off-shelf regions, but biases persist over the shelf area, indicating the need for further enhancements in surface forcing, vertical mixing, and light attenuation.

Here when saying "North Sea operational model" are the authors referring to the Copernicus Marine NWS-MFC forecasting model system? If so, and the increase of

resolution mentioned is the one documented in **Tonani et. al. (2019),** please, refer properly to such work (reference below). If not, please specify which system and resolution increase is here being mentioned.

Tonani, M., Sykes, P., King, R. R., McConnell, N., Péquignet, A.-C., O'Dea, E., Graham, J. A., Polton, J., and Siddorn, J.: The impact of a new high-resolution ocean model on the Met Office North-West European Shelf forecasting system, Ocean Sci., 15, 1133–1158, https://doi.org/10.5194/os-15-1133-2019, 2019.

Authors response: We confirm that the reference pertains to the Copernicus Marine NWS-MFC forecasting model system. The statement has been revised to properly cite Tonani et al. (2019) to acknowledge the documented resolution increase and its impact on model performance.

- **In 257.** Fine spatial resolution in unstructured-grid models allows for the resolution of secondary (transversal) circulation in estuaries and straits **(Ilicak et al. 2021),** thereby improving mixing and enhancing the representation of long-channel changes in stratification, as demonstrated by Haid et al.

The Ilicak et al. 2021 paper nicely illustrates how a high-resolution unstructured grid model is used to enhance the simulation of circulation across the Turkish Strait System that communicate both Mediterranean and Black Seas.

Ilicak, M., Federico, I., Barletta, I., Mutlu, S., Karan, H., Ciliberti, S. A., Clementi, E., Coppini, G., & Pinardi, N. (2021). Modeling of the Turkish Strait System Using a High Resolution Unstructured Grid Ocean Circulation Model. *Journal of Marine Science and Engineering*, *9*(7), 769. https://doi.org/10.3390/jmse9070769

Authors response: The discussion on the added value of unstructured-grid modeling for straits has been extended, incorporating the example from Ilicak et al. (2021) to illustrate its application in simulating circulation across the Turkish Strait System. Additionally, we have included other studies on strait modeling using unstructured-grid approaches to provide a broader perspective on their effectiveness in resolving complex hydrodynamics in these environments.

- **In 266.** Data assimilation in coastal regions presents challenges due to the presence of multiple scales and competing forcings from open boundaries, rivers, and the atmosphere, which are often imperfectly known (Moore and Martin, 2019).

In this point, I would suggest adding specific mention to tides as one of the main challenges for data assimilation. There are many references in the literature to the (absence of) data assimilation in tidal coastal zones. One statement like the following one can be added to the manuscript: **Data assimilation is particularly challenging in tidal environments (especially for meso- and macro-tidal environments; and not so in micro-tidal coastal zones).** -included selected update references...-

**Authors response:** We have extended the discussion to explicitly highlight tides as a key challenge for data assimilation in coastal regions, particularly in meso- and macro-tidal environments. Several references have been added to support this statement, addressing the complexities introduced by tidal dynamics in data assimilation processes.

- Some parts of the manuscript lack citation. This is especially so, for instance in section 2. Unlikely, some other sections of the manuscript provide much more level of references to previous works than Section 2. Below, some options for balancing this different level of citation, including some pertinent reference to address the following points can be:

**In 82.** High-resolution services in the coastal ocean operate at various spatial scales depending on the specific applications and objectives (**Sotillo, 2022**).

This Special Issue (Sotillo, 2022) entitled "ocean modelling in support of operational ocean and coastal services", compiles 11 recent papers on operational coastal services based on high-resolution models. Its citation here can certainly provide readers with insights about scales, objectives and applications, as stated in this sentence with no reference.

Sotillo, M. G. (2022). Ocean Modelling in Support of Operational Ocean and Coastal Services. Journal of Marine Science and Engineering, 10(10), 1482. https://doi.org/10.3390/jmse10101482

**Authors response:** The referenced collection has been added to provide insights into the scales, objectives, and applications of high-resolution coastal ocean modeling.

**In 135.** Accounting for high-resolution atmospheric forcing into coastal models is essential for accurately capturing local meteorological dynamics, including wind patterns, temperature gradients, and precipitation rates. Such detailed atmospheric data drive fundamental processes like heat and momentum fluxes **(García-León et al. 2022),** profoundly influencing coastal hydrodynamics, sediment transport, and ecosystem response.

The proposed García-León et al. work demonstrates that the usage of a new high-resolution atmospheric forcing, together with the update of bulk formulae to compute surface fluxes, have positive impacts across different high-resolution model systems for ports.

García-León, M., Sotillo, M. G., Mestres, M., Espino, M., & Fanjul, E. Á. (2022). Improving Operational Ocean Models for the Spanish Port Authorities: Assessment of the SAMOA Coastal Forecasting Service Upgrades. *Journal of Marine Science and Engineering, 10*(2), 149. https://doi.org/10.3390/jmse10020149.

Authors The discussion on atmospheric forcing has been extended to emphasize its role in coastal modeling, particularly in driving key ocean-atmosphere interactions. Additional references have been included to support the impact of high-resolution atmospheric forcing and improved bulk formulae on model performance.

This is one example, among many others in the literature (authors should fell free to add more), that can support such statement on atmospheric forcing. I would recommend the authors look for some more references to complete this aspect related to the improvement of the atmospheric forcing in coastal high-resolution modeling. Indeed, it would be interesting if in a review paper like this, it is mentioned that atmospheric forcing can be seen today unfortunately as a common limitation for high-resolution coastal modelling. Especially, when coastal modelers are aiming and working on (as stated at the end of this section 2; in 158):

"Simulations at grid resolutions that would sufficiently resolve the coastal sub mesoscale would require horizontal grid resolutions of approximately 1-10 meters in estuaries and 0.1-1 kilometer in coastal shelf domains. However, achieving such high resolutions poses significant computational challenges and resource demands".

But coastal modelers typically can rely only on atmospheric forcing data from national/regional operational services, which have resolutions of around 2-5 km resolution, often being this the best available option (or even lower resolution data when no other alternatives are available). Authors may also link this point on atmospheric forcing limitations with on-going AI initiatives to improve coastal winds.

Authors response: We appreciate the reviewer's insightful comments. We have expanded our discussion on the limitations of atmospheric forcing in high-resolution coastal modeling, emphasizing its role as a key constraint in resolving sub-mesoscale processes. To address this, we have integrated additional references, which highlight the challenges associated with coarse-resolution atmospheric data. Furthermore, we have included a discussion on emerging AI-driven approaches for enhancing coastal wind fields, which offer promising advancements in improving the accuracy of atmospheric forcing in regional models.

**In 149.** The interactions between tidal forcing, river flow and estuarine geometry result in intricate and variable periodic patterns (as shown in **Campuzano et al. 2022 for the Western Iberian Buoyant Plume and in Sotillo et al. 2021 for the whole European Atlantic façade**).

Campuzano et al. and Sotillo et al. works (on the simulation of the Western Iberian Buoyant Plume formed by the contribution of several rivers, and the sensitivity of IBI model to different river forcing data) can provide some illustration of the intricate and variable patterns resulting between river flows, estuarine geometries; and all in regions with marked tidal influence.

Campuzano, F., Santos, F., Simionesei, L., Oliveira, A. R., Olmedo, E., Turiel, A., Fernandes, R., Brito, D., Alba, M., Novellino, A., & Neves, R. (2022). Framework for Improving Land Boundary Conditions in Ocean Regional Products. *Journal of Marine Science and Engineering*, *10*(7), 852. https://doi.org/10.3390/jmse10070852

Sotillo, M. G., Campuzano, F., Guihou, K., Lorente, P., Olmedo, E., Matulka, A., Santos, F., Amo-Baladrón, M. A., & Novellino, A. (2021). River Freshwater Contribution in Operational Ocean Models along the European Atlantic Façade: Impact of a New River Discharge Forcing Data on the CMEMS IBI Regional Model Solution. *Journal of Marine Science and Engineering*, *9*(4), 401. https://doi.org/10.3390/jmse9040401

Authors response: Thank you for the suggestion. We have expanded the section to further illustrate the complex interactions between tidal forcing, river flow, and estuarine geometry. The references to Campuzano et al. (2022) and Sotillo et al. (2021) have been incorporated to support this discussion.

- Some other references that may enhance the proposed introductory/review scope of the paper may be the following ones:

**In 197**. The use of HFR networks has become an essential element of coastal ocean observing systems, contributing to high-level coastal services (Stanev et al., 2016; Rubio et al., 2017; **Reyes et al., 2022**)

I would suggest including this more recent review work of Reyes et al. on existing HFR data multidisciplinary science-based applications in the Mediterranean Sea, primarily focused on meeting end-user and science-driven requirements, addressing regional challenges in maritime safety, extreme hazards and environmental transport processes.

Reyes, E., Aguiar, E., Bendoni, M., Berta, M., Brandini, C., Cáceres-Euse, A., Capodici, F., Cardin, V., Cianelli, D., Ciraolo, G., Corgnati, L., Dadić, V., Doronzo, B., Drago, A., Dumas, D., Falco, P., Fattorini, M., Fernandes, M. J., Gauci, A., Gómez, R., Griffa, A., Guérin, C.-A., Hernández-Carrasco, I., Hernández-Lasheras, J., Ličer, M., Lorente, P., Magaldi, M. G., Mantovani, C., Mihanović, H., Molcard, A., Mourre, B., Révelard, A., Reyes-Suárez, C., Saviano, S., Sciascia, R., Taddei, S., Tintoré, J., Toledo, Y., Uttieri, M., Vilibić, I., Zambianchi, E., and Orfila, A.: Coastal high-frequency radars in the Mediterranean – Part 2: Applications in support of science priorities and societal needs, Ocean Sci., 18, 797–837, https://doi.org/10.5194/os-18-797-2022, 2022.

Authors response: Thank you for the suggestion. The reference to Reyes et al. (2022) has been added to further support the discussion on the role of HFR networks in coastal ocean observing systems and their applications in addressing regional challenges in the Mediterranean Sea.

- Minor (typo) Points.

In 15 typo: "introduce key"

In 17 typo: "for the"

 The typos have been corrected. Thank you

---

## Author Comment (AC3)

**EC1**: , Kirsten Wilmer-Becker, 19 Feb 2025  reply

Dear Dr. Wilmer-Becker,

We would like to thank you for your careful reading of our manuscript and for your constructive comments. We appreciate the suggestions, which helped us improve the clarity and completeness of the text. Below, we provide a point-by-point response to your remarks. Revisions made in the manuscript are highlighted accordingly.

We thank you once again for your helpful feedback and for the opportunity to revise our manuscript.

Best regards,

Joanna Staneva

This paper highlights some important aspects that should be considered when applying a numerical model to the coastal ocean, particularly in an operational context. While this style of paper is not new, and there have been similar contributions to the literature in the past, the state-of-the-art in coastal ocean modelling methodologies does advance, and it is appropriate that the appraisal of those methodologies should similarly periodically advance. Additionally, there are a broad range of issues that require consideration in the coastal ocean from a modelling perspective, and most existing reviews of coastal modelling techniques focus on a subset of this range. This manuscript is no different, choosing to focus on spatial scales & processes, observations, nesting, unstructured approaches, and observing system experiments. These aspects are indeed relevant for coastal zone modelling, and consideration of this subset does not detract from its general relevance in my view. While the topics under consideration in the manuscript may not come as revelations to modellers well versed in coastal applications, they are central to producing good coastal models, and are a timely reminder that these aspects should receive close attention when building a coastal zone model. The speculation in the summary around how contemporary trends, driven by coastal necessities, may influence future applications is a good synthesis of where coastal modelling is heading. As such, I think this manuscript is a worthy addition to the literature and I recommend publication with some minor alterations.

Authors' response: Thank you. We appreciate the constructive evaluation and take note of the positive assessment regarding the scope and relevance of the manuscript. The intention here is indeed to present a focused review of selected elements central to high-resolution coastal modelling, particularly in operational frameworks. We have addressed the specific suggestions in the following responses.

In Section 3.1, novel observational platforms are considered, with HF radar and ADCPs in particular singled out for attention. I think it may also be worthwhile to make explicit mention of slocum gliders here (i.e., prior to its brief mention in Section 4). These autonomous underwater vehicles can host a wide array of instrumentation, and deliver high spatial and temporal resolution observations, especially if repeat transects are programmed. Similarly, while SWOT is a step forward in terms of remotely sensing the coastal ocean, the geostationary Himawari-8 satellite is similarly a step forward, delivering up to 500m and 10 minute resolution data, and maybe also worth a mention. I think any current review of observations should probably include these contemporary platforms.

Authors' response: Thank you. We agree that including additional observation platforms is relevant. The revised manuscript now explicitly refers to gliders and the Himawari-8 satellite, with a brief explanation of their application in the context of high-resolution coastal observations and provided additional references. We also updated the text about SWOT applications in the coastal ocean.

Some statements are made that would carry more weight if additional references or examples were given, particularly in Section 2; e.g., paragraph starting line 130 regarding small spatial scales, line 237-238, paragraph starting line 239, with perhaps additional examples outside the Baltic, line 249-250 for riverine input methods, and in general where qualitative statements are made throughout.

Authors' response: Thank you. We have reviewed the mentioned sections and added references where appropriate. These include examples from different regions, as well as studies that illustrate the methods discussed. Please also refer to our detailed responses to both reviewers for specific updates and newly cited literature. For the riverine input methods, we have added a dedicated subsection discussing the role of river discharge in coastal dynamics and modelling, supported by recent high-resolution studies.

Section 3.3. The type of open boundary applied to downscaled models is key to a good solution free from specification error. OBCs are generally not well transportable across applications, and require some application-specific tuning. An ocean model with a large suite of OBCs is advantageous when solving coastal ocean problems. I think the manuscript could be strengthened with some commentary around open boundary conditions, perhaps an elaboration of lines 151-154 with references.

Authors' response: Thank you. We have expanded the discussion in Section 3.3 to include a short paragraph on open boundary conditions. We refer to common challenges and the importance of case-specific configuration. Additional references have been included.

Line 250: 'Unstructured-grid models, with their ability to employ higher-order spatial discretizations' – this isn't strictly true as unstructured models more commonly employ

lower order momentum and tracer advection owing to their irregular grid and awkward interpolations required to achieve higher order. They can, however, provide superior resolution placement and transition, allowing better dynamic representation in coastal and estuarine environments.

Authors' response: Thank you for this remark. Thank you for this constructive remark. We agree that unstructured-grid models typically employ lower-order discretization due to interpolation challenges on irregular meshes. The revised sentence now reflects this more accurately, while emphasizing the strength of such models in resolving complex dynamics through flexible resolution placement.

Paragraph starting line 260. Grid generators tailored for the specific requirements of the unstructured numerical core are starting to appear, e.g., JIGSAW (Engwirda, 2017, Geosci. Model Dev, 10 (6), p. 2117). This package creates high quality meshes that are an orthogonal, well centred centroidal Voronoi tessellation, that, for example, conform to the numerical requirements of TRiSK. This package is also seeing uptake in other cores, often with a front-end API attached (e.g., OCSMesh). These numerics-tailored mesh generators are in contrast to older meshing packages, e.g., John Shewchuk's TRIANGLE, which is a general-purpose triangulation package which has been used by modellers in the past, and is not specifically tailored to solving the Navier Stokes on a mesh. Perhaps the progress of JIGSAW style triangulators for more objective mesh generation could be mentioned.

Authors' response: This suggestion has been addressed by adding a reference to the JIGSAW mesh generator and a short note on its relevance for generating meshes tailored to specific numerical schemes.

Line 152: 'Unlike global models that can operate with open boundaries...' Should this be '.... without open boundaries'?

Authors' response: We agree and have corrected the sentence to: "*Unlike global models that can operate without open boundaries, regional and coastal models require well-defined lateral boundary conditions.*"

Section 3.2, Table 1: This is awkward – the list is good, but I think the coastal unstructured COMPAS model developed by CSIRO (Herzfeld et al, 2020, https://doi.org/10.1016/j.ocemod.2020.101599 ), or global MPAS developed by LANL (Ringler et al., 2013  10.1016/j.ocemod.2013.04.010) could also be a worthy addition. Awkward because I'm beating my own drum here with COMPAS. However, these models are based on the TRiSK numerics which is one of the few numerical cores that operates unstructured with finite volume on a C-grid (in this case Voronoi tessellations) without generating spurious modes that require suppression to control. The core also has other desirable properties that merit its inclusion. Ultimately the authors call though.

Authors' response: Thank you for the helpful suggestion. We agree with the relevance of both models. COMPAS and MPAS have now been added to Table 1 as unstructured finite-volume models, and the corresponding references (Herzfeld et al., 2020; Ringler et al., 2013) are included in the revised manuscript.

---

## Author Response (AR2)

Dear Editors,

Thank you very much for your careful proof-editing and helpful comments.

We have revised the manuscript accordingly and addressed all points raised. Redundancies and repetitions in the text have been removed, and the reference list has been thoroughly reviewed and corrected, including all DOIs and links.

Please find our line-by-line responses below this letter (in blue).

We appreciate your support and look forward to the next steps in the publication process.

Best regards,

Joanna Staneva

The manuscript is of excellent quality and does only need a few updates to be ready for publication:

Thank you.

Some reptition in the introduction, in line 38-42 and 56-61. Please rephrase/consolidate or remove.

Text has been revised, and repeating phrases have been removed. Thank you.

Line 274. The given link is no longer safe to access (no https) and has been archived. An alternative link could be used if useful: https://forge.nemo-ocean.eu/nemo. Please check.

The link has been updated in references. In the main text correct reference has been provided.

Some repetition in line 303-306 compared to line 307-315. Please rephrase/consolidate or remove

Redundant text has been removed. Thank you

The references list requires some checks and revisions:

Thank you for the thoughtful review. The references have been carefully checked, and all links and DOIs have been verified and corrected.

Line 496 please add full link: https://doi.org/10.1029/CO004p0001

Full link has been added. Thank you.

Line 501 link not working (replace hyphens with forward slashes)

Done.

Line 502 link not working (last part: "<425::AID-FLD847>3.0.CO;2-D" is not part of link)

Done.

Line 505 link not working (remove two dots at the end link)

Done.

Line 510 link not working (remove dot at the end of link)

Done.

Line 513 please add full link: https://doi.org/10.3389/fmars.2023.1105626

Done.

Line 515/516 link given is not complete, please update

Done.

Line 522 link not working (remove dot at the end of link).

Done.

Line 524 link not working (remove dot at the end of link).

Done.

Line 530 link not correct, please check

Link corrected.

Line 535 please update paper title/compare link

Paper title has been updated. Paper link updated.

Line 539 link not working (remove dot at the end of link).

Done.

Line 541 link not working (remove dot at the end of link).

Done.

Line 543 doi not working, please use

https://cordis.europa.eu/project/id/101133911/results

The link has been added. In addition the DOI has been updated

Line 547 link not working (remove dot at the end of link).

Done.

Line 552 link not working (remove dot at the end of link).

Done.

Line 555 link not working (remove dot at the end of link).

Done.

Line 557 link not working (remove dot at the end of link).

Done.

Line 560 link not working (remove comma at the end of link).

Done.

Line 567/568 - doi incorrect, cannot locate the paper

DOI link  has been corrected. Thank you.

Line 571 link not working (remove dot at the end of link).

Done.

Line 576 link not working (remove dot at the end of link).

Done.

Line 578 link not working (remove dot at the end of link).

Done.

Line 583 link not working (remove dot at the end of link).

Done.

Line 587 link not working (remove dot at the end of link).

Done.

Line 593 link not working (remove dot at the end of link).

Done.

Line 594 link not working, please update to: https://sp.copernicus.org/preprints/sp-2024-9/

The correct link has been updated.

Line 598 link not working (remove dot at the end of link).

Done.

Line 605 link not working (remove dot at the end of link).

Done.

Line 610 link not working (remove dot at the end of link).

Done.

Line 612 link not working (remove dot at the end of link).

Done.

Line 614 link not working (remove dot at the end of link).

Done.

Line 616: please udpate paper title and current doi with: https://a.tellusjournals.se/articles/10.3402/tellusa.v66.21640

Link updated. Thank you.

Line 618 link not working (remove dot at the end of link).

Done.

Line 625 link not working (remove dot at the end of link).

Done.

Line 627 link not working (remove dot at the end of link).

Done.

Line 633 link not working (remove dot at the end of link).

Done.

Line 635 link not working (remove dot at the end of link).

Done.

Line 638 please use link: https://doi.org/10.1016/j.csr.2021.104582

Link has been updated. Thank you.

Line 641 link not working (remove dot at the end of link).

Done.

Line 651 link not working (remove comma at the end of link).

Done.

Line 653 please add break to position next publication correctly. Link address https://doi.org/10.1016/j.ocemod.2013.04.010 can be used here.

Done.

Line 654 Link not working as "marine-122414-033913" missing from link address.

Done.

Line 661 link not working (remove dot at the end of link).

Done.

Line 664 link not working (remove dot at the end of link).

Done.

Line 668 link not working (remove dot at the end of link).

Done.

Line 670 link not working (remove dot at the end of link).

Done.

Line 675 link not working (remove dot at the end of link).

Done.

Line 697/680 couldn't verify doi number. Please check.

DOI has been updated. Thank you

Line 685 link not working (remove comma at the end of link).

Done.

Line 688 link not working (remove dot at the end of link).

Done.

Line 690 link not working (remove dot at the end of link).

Done.

Line 692 link not working (remove dot at the end of link).

Done.

Line 695 link not working (remove dot at the end of link).

Done.

Line 704 please replace doi with link: https://doi.org/10.3389/fmars.2021.644525

DOI has been replaced with the link. Thank you.

Line 710 link not working (remove dot at the end of link).

Done.

Line 721 link not working (remove dot at the end of link).

Done.

Line 724 link not working (remove dot at the end of link).

Done.

Line 727 link not working (remove dot at the end of link).

Done.

Line 733 link not working (remove dot at the end of link).

Done.

Line 735 option to add full link: https://doi.org/10.1175/BAMS-D-19-0155.1

Full link has been used. Thank you!

Line 737 link not working (remove dot at the end of link).

Done.

Line 739 link not working (remove dot at the end of link).

Done.

Line 741 link not working (remove dot at the end of link).

Done.